# Benchmarking Deep Learning Interpretability in Time Series Predictions

**Aya Abdelsalam Ismail, Mohamed Gunady, Héctor Corrada Bravo**\*, **Soheil Feizi** \*
{asalam,mgunady,sfeizi}@cs.umd.edu, hcorrada@umiacs.umd.edu
Department of Computer Science, University of Maryland

## Abstract

Saliency methods are used extensively to highlight the importance of input features in model predictions. These methods are mostly used in vision and language tasks, and their applications to time series data is relatively unexplored. In this paper, we set out to extensively compare the performance of various saliency-based interpretability methods across diverse neural architectures, including Recurrent Neural Network, Temporal Convolutional Networks, and Transformers in a new benchmark [†] of synthetic time series data. We propose and report multiple metrics to empirically evaluate the performance of saliency methods for detecting feature importance over time using both precision (i.e., whether identified features contain meaningful signals) and recall (i.e., the number of features with signal identified as important). Through several experiments, we show that (i) in general, network architectures and saliency methods fail to reliably and accurately identify feature importance over time in time series data, (ii) this failure is mainly due to the conflation of time and feature domains, and (iii) the quality of saliency maps can be improved substantially by using our proposed two-step temporal saliency rescaling (TSR) approach that first calculates the importance of each time step before calculating the importance of each feature at a time step.

## 1 Introduction

As the use of Machine Learning models increases in various domains [1, 2], the need for reliable model explanations is crucial [3, 4]. This need has resulted in the development of numerous interpretability methods that estimate feature importance [5–13]. As opposed to the task of understanding the prediction performance of a model, measuring and understanding the performance of interpretability methods is challenging [14–18] since there is no ground truth to use for such comparisons. For instance, while one could identify sets of informative features for a specific task a priori, models may not necessarily have to draw information from these features to make accurate predictions. In multivariate time series data, these challenges are even more profound since we cannot rely on human perception as one would when visualizing interpretations by overlaying saliency maps over images or when highlighting relevant words in a sentence.

In this work, we compare the performance of different interpretability methods both perturbation-based and gradient-based methods, across diverse neural architectures including Recurrent Neural Network, Temporal Convolutional Networks, and Transformers when applied to the classification of multivariate time series. We quantify the performance of every (architectures, estimator) pair for time series data in a systematic way. We design and generate multiple synthetic datasets to capture different temporal-spatial aspects (e.g., Figure 1). Saliency methods must be able to distinguish important and non-important features at a given time, and capture changes in the importance of

---

[†]Code: https://github.com/ayaabdelsalam91/TS-Interpretability-Benchmark

features over time. The positions of informative features in our synthetic datasets are known a priori (colored boxes in Figure 1); however, the model might not need *all* informative features to make a prediction. To identify features *needed* by the model, we progressively mask the features identified as important by each interpretability method and measure the accuracy degradation of the trained model. We then calculate the precision and recall for (architectures, estimator) pairs at different masks by comparing them to the known set of informative features.

Based on our extensive experiments, we report the following observations: (i) feature importance estimators that produce high-quality saliency maps in images often fail to provide similar high-quality interpretation in time series data, (ii) saliency methods tend to fail to distinguish important vs. non-important features in a given time step; if a feature in a given time is assigned to high saliency, then almost all other features in that time step tend to have high saliency regardless of their actual values, (iii) model architectures have significant effects on the quality of saliency maps.

After the aforementioned analysis and to improve the quality of saliency methods in time series data, we propose a two-step **T**emporal **S**aliency **R**escaling (**TSR**) approach that can be used on top of any existing saliency method adapting it to time series data. Briefly, the approach works as follows: (a) we first calculate the *time-relevance score* for each time by computing the total change in saliency values if that time step is masked; then (b) in each time-step whose time-relevance score is above a certain threshold, we calculate the *feature-relevance score* for each feature by computing the total change in saliency values if that feature is masked. The final (time, feature) importance score is the product of associated time and feature relevance scores. This approach substantially improves the quality of saliency maps produced by various methods when applied to time series data. Figure 4 shows the initial performance of multiple methods, while Figure 5 shows their performance coupled with our proposed TSR method.

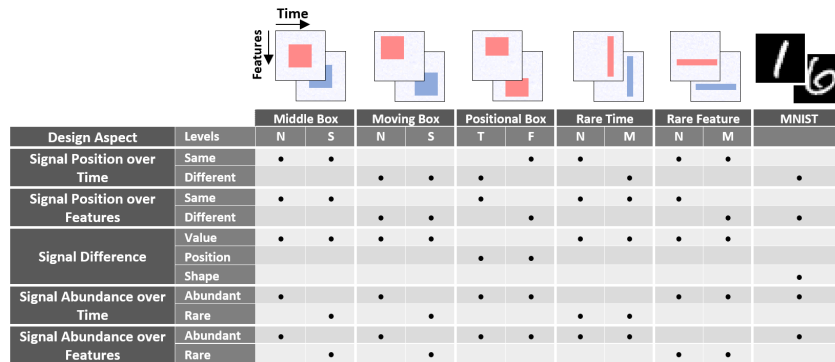

Figure 1: Different evaluation datasets used for benchmarking saliency methods. Some datasets have multiple variations shown as sub-levels. N/S: normal and small shapes, T/F: temporal and feature positions, M: moving shape. All datasets are trained for binary classification, except MNIST. Examples are shown above each dataset, where dark red/blue shapes represent informative features.

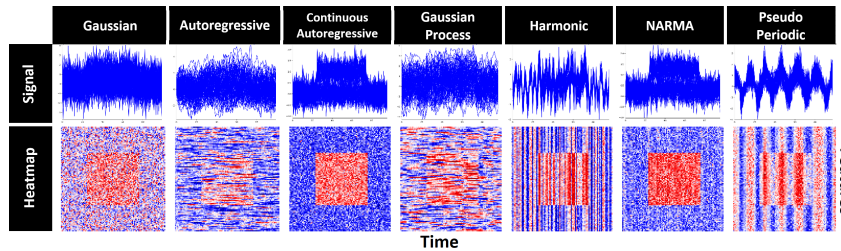

Figure 2: Middle box dataset generated by different time series processes. The first row shows how each feature changes over time when independently sampled from time series processes. The bottom row corresponds to the heatmap of each sample where red represents informative features.

## 2  Background and Related Work

The interest in interpretability resulted in several diverse lines of research, all with a common goal of understanding how a network makes a prediction. [19–23] focus on making neural models more interpretable. [24, 9, 11, 6, 7, 25] estimate the importance of an input feature for a specified output. Kim et al. [26] provides an interpretation in terms of human concepts. One key question is whether or not interpretability methods are reliable. Kindermans et al. [17] shows that the explanation can be manipulated by transformations that do not affect the decision-making process. Ghorbani et al. [15] introduces an adversarial attack that changes the interpretation without changing the prediction. Adebayo et al. [16] measures changes in the attribute when randomizing model parameters or labels.

Similar to our line of work, modification-based evaluation methods [27–29] involves: applying saliency method, ranking features according to the saliency values, recursively eliminating higher ranked features and measure degradation to the trained model accuracy. Hooker et al. [14] proposes retraining the model after feature elimination.

Recent work [23, 30, 31] have identified some limitations in time series interpretability. We provide the first benchmark that systematically evaluates different saliency methods across multiple neural architectures in a multivariate time series setting, identifies common limitations, and proposes a solution to adapt existing methods to time series.

### 2.1  Saliency Methods

We compare popular backpropagation-based and perturbation based post-hoc saliency methods; each method provides feature importance, or relevance, at a given time step to each input feature. All methods are compared with **random assignment** as a baseline control.

In this benchmark, the following saliency methods[†] are included:

- **Gradient-based:** *Gradient (GRAD)* [5] the gradient of the output with respect to the input. *Integrated Gradients (IG)* [9] the average gradient while input changes from a non-informative reference point. *SmoothGrad (SG)* [10] the gradient is computed $n$ times, adding noise to the input each time. *DeepLIFT (DL)* [11] defines a reference point, relevance is the difference between the activation of each neuron to its reference activation. *Gradient SHAP (GS)* [12] adds noise to each input, selects a point along the path between a reference point and input, and computes the gradient of outputs with respect to those points. *Deep SHAP (DeepLIFT + Shapley values) (DLS)* [12] takes a distribution of baselines computes the attribution for each input-baseline pair and averages the resulting attributions per input.

- **Perturbation-based:** *Feature Occlusion (FO)* [24] computes attribution as the difference in output after replacing each contiguous region with a given baseline. For time series we considered continuous regions as features with same time step or multiple time steps grouped together. *Feature Ablation (FA)* [32] computes attribution as the difference in output after replacing each feature with a baseline. Input features can also be grouped and ablated together rather than individually. *Feature permutation (FP)* [33] randomly permutes the feature value individually, within a batch and computes the change in output as a result of this modification.

- **Other:** *Shapley Value Sampling (SVS)* [34] an approximation of Shapley values that involves sampling some random permutations of the input features and average the marginal contribution of features based the differences on these permutations.

### 2.2  Neural Net Architectures

In this benchmark, we consider 3 main neural architectures groups; Recurrent networks, Convolution neural networks (CNN) and Transformer. For each group we investigate a subset of models that are commonly used for time series data. Recurrent models include: **LSTM** [35] and **LSTM with Input-Cell Attention** [23] a variant of LSTM with that attends to inputs from different time steps. For CNN, **Temporal Convolutional Network (TCN)** [36–38] a CNN that handles long sequence time series. Finally, we consider the original **Transformers** [39] implementation.

---

[†]Captum implementation of different methods was used.

## 3  Problem Definition

We study a time series classification problem where all time steps contribute to making the final output; labels are available after the last time step. In this setting, a network takes multivariate time series input $X = [x_1, \ldots, x_T] \in \mathbb{R}^{N \times T}$, where $T$ is the number of time steps and $N$ is the number of features. Let $x_{i,t}$ be the input feature $i$ at time $t$. Similarly, let $X_{:,t} \in \mathbb{R}^N$ and $X_{i,:} \in \mathbb{R}^T$ be the feature vector at time $t$, and the time vector for feature $i$, respectively. The network produces an output $S(X) = [S_1(X), ..., S_C(X)]$, where $C$ is the total number of classes (i.e. outputs). Given a target class $c$, the saliency method finds the relevance $R(X) \in \mathbb{R}^{N \times T}$ which assigns relevance scores $R_{i,t}(X)$ for input feature $i$ at time $t$.

## 4  Benchmark Design and Evaluation Metrics

### 4.1  Dataset Design

Since evaluating interpretability through saliency maps in multivariate time series datasets is nontrivial, we design multiple synthetic datasets where we can control and examine different design aspects that emerge in typical time series datasets. We extend the synthetic data proposed by Ismail et al. [23] for binary classification. We consider how the discriminating signal is distributed over both time and feature axes, reflecting the importance of time and feature dimensions separately. We also examine how the signal is distributed between classes: difference in value, position, or shape. Additionally, we modify the classification difficulty by decreasing the number of informative features (reducing feature redundancy), i.e., *small box datasets*. Along with synthetic datasets, we included MNIST as a multivariate time series as a more general case (treating one of the image axes as time). Different dataset combinations are shown in Figure 1.

Each synthetic dataset is generated by seven different processes as shown in Figure 2, giving a total of 70 datasets. Each feature is independently sampled from either: (a) Gaussian with zero mean and unit variance. (b) Independent sequences of a standard autoregressive time series with Gaussian noise. (c) A standard continuous autoregressive time series with Gaussian noise. (d) Sampled according to a Gaussian Process mixture model. (e) Nonuniformly sampled from a harmonic function. (f) Sequences of standard non–linear autoregressive moving average (NARMA) time series with Gaussian noise. (g) Nonuniformly sampled from a pseudo period function with Gaussian noise. Informative features are then highlighted by the addition of a constant $\mu$ to positive class and subtraction of $\mu$ from negative class (unless specified, $\mu = 1$); the embedding size for each sample is $N = 50$, and the number of time steps is $T = 50$. Figures throughout the paper show data generated as Gaussian noise unless otherwise specified. Further details are provided in the supplementary material.

### 4.2  Feature Importance Identification

Modification-based evaluation metrics [27–29] have two main issues. First, they assume that feature ranking based on saliency faithfully represents feature importance. Consider the saliency distributions shown in Figure 3. Saliency decays exponentially with feature ranking, meaning that features that are closely ranked might have substantially different saliency values. A second issue, as discussed by Hooker et al. [14], is that eliminating features changes the test data distribution violating the assumption that both training and testing data are independent and identically distributed (i.i.d.). Hence, model accuracy degradation may be a result of changing data distribution rather than removing salient features. In our synthetic dataset benchmark, we address these two issues by the following:

- Sort relevance $R(X)$, so that $R_e(x_{i,t})$ is the $e^{th}$ element in ordered set $\{R_e(x_{i,t})\}_{e=1}^{T \times N}$.

- Find top $k$ relevant features in the order set such that $\frac{\sum_{e=1}^{k} R_e(x_{i,t})}{\sum_{i=1,t=1}^{N,T} R(x_{i,t})} \approx d$ (where $d$ is a pre-determined percentage).

- Replace $x_{i,t}$, where $R(x_{i,t}) \in \{R_e(x_{i,t})\}_{e=1}^{k}$ with the original distribution (known since this is a synthetic dataset).

- Calculate the drop in model accuracy after the masking, this is repeated at different values of $d = [0, 10, \ldots, 100]$.

We address the first issue by removing features that represent a certain percentage of the overall saliency rather than removing a constant number of features. Since we are using synthetic data and masking using the original data distribution, we are not violating i.i.d. assumptions.

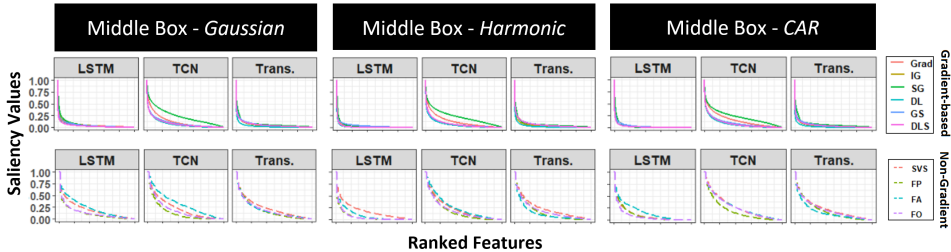

Figure 3: The saliency distribution of ranked features produced by different saliency methods for three variations of the Middle Box dataset (Gaussian, Harmonic, Continous Autoregressive (CAR)). Top row shows gradient-based saliency methods while bottom row shows the rest.

### 4.3  Performance Evaluation Metrics

Masking salient features can result in (a) a steep drop in accuracy, meaning that the removed feature is *necessary* for a correct prediction or (b) unchanged accuracy. The latter may result from the saliency method incorrectly identifying the feature as important, or that the removal of that feature is not *sufficient* for the model to behave incorrectly. Some neural architectures tend to use more feature information when making a prediction (i.e., have more recall in terms of importance); this may be the desired behavior in many time series applications where importance changes over time, and the goal of using an interpretability measure is to *detect* all relevant features across time. On the other hand, in some situations, where sparse explanations are preferred, then this behavior may not be appropriate. This in mind, one should not compare saliency methods solely on the loss of accuracy after masking. Instead, we should look into features identified as salient and answer the following questions: (1) ***Are all features identified as salient informative?*** *(precision)* (2) ***Was the saliency method able to identify all informative features?*** *(recall)*

We choice to report the *weighted* precision and recall of each *(neural architecture, saliency method)* pair, since, the saliency value varies dramatically across features Figure 3 (detailed calculations are available in the supplementary material).

Through our experiments, we report area under the precision curve (AUP), the area under the recall curve (AUR), and area under precision and recall (AUPR). The curves are calculated by the precision/recall values at different levels of degradation. We also consider feature/time precision and recall (a feature is considered informative if it has information at any time step and vice versa). For the random baseline, we stochastically select a saliency method then permute the saliency values producing arbitrary ranking.

## 5  Saliency Methods Fail in Time Series Data

Due to space limitations, only a subset of the results is reported below; the full set is available in the supplementary material. The results reported in the following section are for models that produce accuracy above 95% in the classification task.

### 5.1  Saliency Map Quality

Consider synthetic examples in Figure 4; given that the model was able to classify all the samples correctly, one would expect a saliency method to highlight only informative features. However, we find that for the *Middle Box* and *Rare Feature* datasets, many different (neural architecture, saliency method) pairs are unable to identify informative features. For *Rare time*, methods identify the correct time steps but are unable to distinguish informative features within those times. Similarly, methods were not able to provide quality saliency maps produced for the multivariate time series MNIST digit. Overall most (neural architecture, saliency method) pairs fail to identify importance over time.

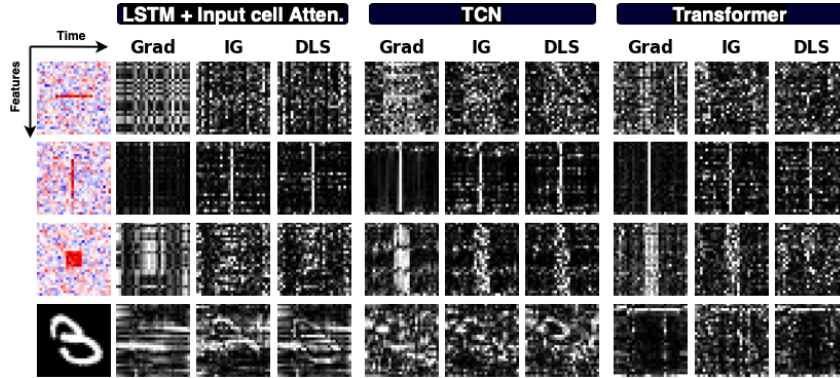

Figure 4: Saliency maps produced by Grad, Integrated Gradients, and DeepSHAP for 3 different models on synthetic data and time series MNIST (white represents high saliency). Saliency seems to highlight the correct time step in some cases but fails to identify informative features in a given time.

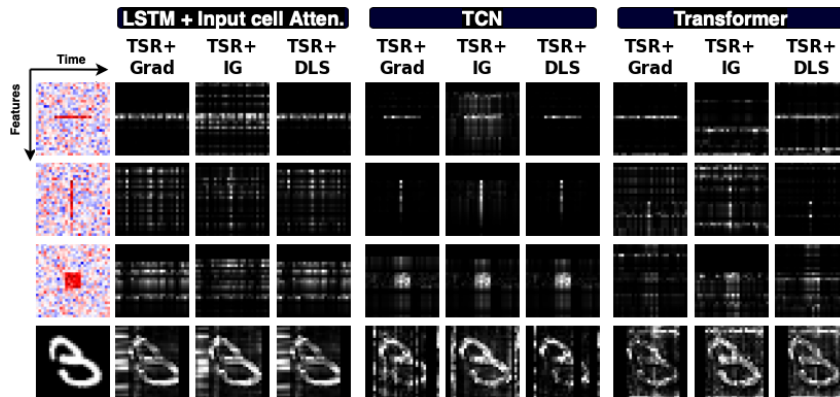

Figure 5: Saliency maps when applying the proposed Temporal Saliency Rescaling (TSR) approach.

## 5.2 Saliency Methods versus Random Ranking

Here we look into distinctions between each saliency method and a random ranking baseline. The effect of masking salient features on the model accuracy is shown in Figure 6. In a given panel, the leftmost curve indicates the saliency method that highlights a small number of features that impact accuracy severely (if correct, this method should have high precision); the rightmost curve indicates the saliency method that highlights a large number of features that impact accuracy severely (if correct, this method should show high recall).

**Model Accuracy Drop**

We were unable to identify a consistent trend for saliency methods across all neural architectures throughout experiments. Instead, saliency methods for a given architecture behave similarly across datasets. E.g., in TCN Grad and SmoothGrad had steepest accuracy drop across all datasets while LSTM showed no clear distinction between random assignment and non-random saliency method curves (this means that LSTM is very hard to interpret regardless of the saliency method used as [23]) Variance in performance between methods can be explained by the dataset itself rather than the methods. E.g., the *Moving box* dataset showed minimal variance across all methods, while *Rare time* dataset showed the highest.

**Precision and Recall**

Looking at precision and recall distribution box plots Figure 7 (the precision and recall graphs per dataset are available in the supplementary materials), we observe the following: (a) Model architecture has the largest effect on precision and recall. (b) Results do not show clear distinctions between

saliency methods. (c) Methods can identify informative time steps while fail to identify informative features; AUPR in the time domain (second-row Figure 7) is higher than that in the feature domain (third-row Figure 7). (d) Methods identify most features in an informative time step as salient, AUR in feature domain is very high while having very low AUP. This is consistent with what we see in Figure 4, where all features in informative time steps are highlighted regardless of there actual values. (e) Looking at AUP, AUR, and AUPR values, we find that the steepness in accuracy drop depends on the dataset. A steep drop in model accuracy does not indicate that a saliency method is correctly identifying features used by the model since, in most cases, saliency methods with leftmost curves in Figure 6 have the lowest precision and recall values.

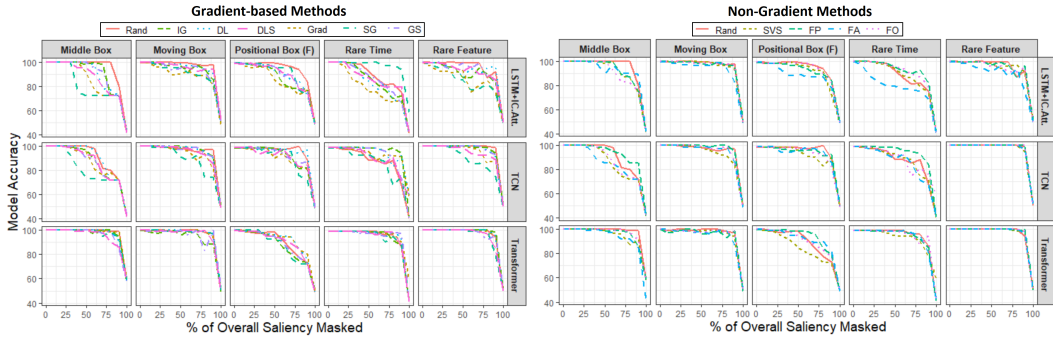

Figure 6: The effect of masking features identified as salient by different methods against a random baseline. Gradient-based and non-gradient based saliency methods are shown in the left and right plots, respectively. The rate of accuracy drop is not consistent; in many cases there is not much improvement over random baseline.

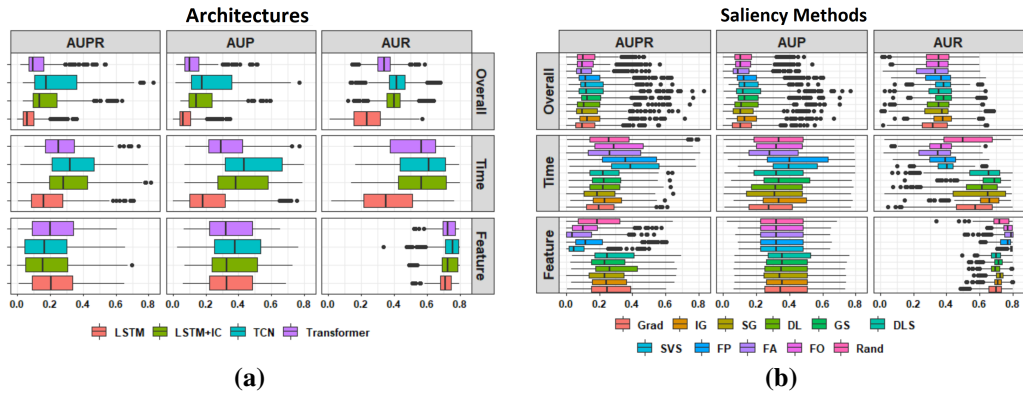

Figure 7: Precision and Recall distribution box plots, the top row represents overall Precision/Recall, while the second two rows show Precision/Recall distribution on time and feature axes (a) Distribution across architectures. (b) Distribution across saliency methods.

## 6 Saliency Maps for Images versus Multivariate Time Series

Since saliency methods are commonly evaluated on images, we compare the saliency maps produced from models like CNN, which fit images, to the maps produced by temporal models like TCN, over our evaluation datasets by treating the complete multivariate time series as an image. Figure 8(a) shows two examples of such saliency maps. The maps produced by CNN can distinguish informative pixels corresponding to informative features in informative time steps. However, maps produced from TCN fall short in distinguishing important features within a given time step. Looking at the saliency distribution of gradients for each model, stratified by the category of each pixel with respect to its importance in both time and feature axes; we find that CNN correctly assigns higher saliency values to pixels with information in both feature and time axes compared to the other categories, which is not the case with TCN, that is biased in the time direction. That observation supports the

conclusion that even though most saliency methods we examine work for images, they generally fail for multivariate time series. It should be noted that this conclusion should not be misinterpreted as a suggestion to treat time series as images (in many cases this is not possible due to the decrease in model performance and increase in dimensionality).

Finally, we examine the effect of reshaping a multivariate time series into univariate or bivariate time series. Figure 8 (b) shows a few examples of saliency maps produced by the various treatment approaches of the same sample (images for CNN, uni, bi, multivariate time series for TCN). One can see that CNN and univariate TCN produce interpretable maps, while the maps for the bivariate and multivariate TCN are harder to interpret. That is due to the failure of these methods to distinguish informative features within informative time steps, but rather focusing more on highlighting informative time steps.

These observations suggest that saliency maps fail when feature and time domains are conflated. When the input is represented solely on the feature domain (as is the case of CNN), saliency maps are relatively accurate. When the input is represented solely on the time domain, maps are also accurate. However, when feature and time domains are both present, the saliency maps across these domains are conflated, leading to poor behavior. This observation motivates our proposed method to adapt existing saliency methods to multivariate time series data.

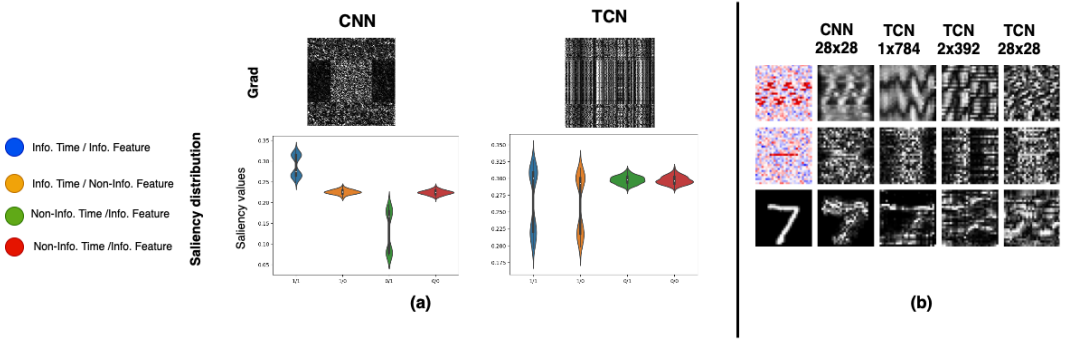

Figure 8: **(a)** Saliency maps and distribution produced by CNN versus TCN for *Middle Box*. **(b)** Saliency Maps for samples treated as image (CNN) vs. uni-, bi- or multi-variate time series (TCN).

# 7  Temporal Saliency Rescaling

From the results presented in previous sections, we conclude that most saliency methods identify informative time steps successfully while they fail in identifying feature importance in those time steps. In this section, we propose a method that can be used on top of any generic interpretation method to boost its performance in time series applications. The key idea is to decouple the (time,feature) importance scores to time and feature relevance scores using a two-step procedure called **T**emporal **S**aliency **R**escaling (**TSR**). In the first step, we calculate the *time-relevance score* for each time by computing the total change in saliency values if that time step is masked. Based on our experiments presented in the last sections, many existing interpretation methods would provide reliable time-relevance scores. In the second step, in each time-step whose time-relevance score is above a certain threshold $\alpha$, we compute the *feature-relevance score* for each feature by computing the total change in saliency values if that feature is masked. By choosing a proper value for $\alpha$, the second step can be performed in a few highly-relevant time steps to reduce the overall computational complexity of the method. Then, the final (time, feature) importance score is the product of associated time and feature relevance scores. The method is formally presented in Algorithm 1.

Figure 5 shows updated saliency maps when applying **TSR** on the same examples in Figures 4. There is a definite improvement in saliency quality across different architectures and interpretability methods except for SmoothGrad; this is probably because SmoothGrad adds noise to gradients, and using a noisy gradient as a baseline may not be appropriate. Table 1 shows the performance of **TSR** with simple Gradient compared to some standard saliency method on the benchmark metrics described in Section 4. **TSR + Grad** outpreforms other methods on all metrics.

**Algorithm 1:** Temporal Saliency Rescaling (TSR)

---

**Given:** input $X$, a baseline interpretation method $R(.)$
**Output:** TSR interpretation method $R^{TSR}(.)$

**for** $t \leftarrow 0$ **to** $T$ **do**
    Mask all features at time $t$: $\overline{X}_{:,t} = 0$, otherwise $\overline{X} = X$;
    Compute Time-Relevance Score $\Delta_t^{time} = \sum_{i,t} |R_{i,t}(X) - R_{i,t}(\overline{X})|$;

**for** $t \leftarrow 0$ **to** $T$ **do**
    **for** $i \leftarrow 0$ **to** $N$ **do**
        **if** $\Delta_t^{time} > \alpha$ **then**
            Mask feature $i$ at time $t$: $\overline{X}_{i,:} = 0$, otherwise $\overline{X} = X$;
            Compute Feature-Relevance Score $\Delta_i^{feature} = \sum_{i,t} |R_{i,t}(X) - R_{i,t}(\overline{X})|$;
        **else**
            Feature-Relevance Score $\Delta_i^{feature} = 0$;
        Compute (time,feature) importance score $R_{i,t}^{TSR} = \Delta_i^{feature} \times \Delta_t^{time}$ ;

---

The proposed rescaling approach improves the ability of saliency methods to capture feature importance over time but significantly increases the computational cost of producing a saliency map. Other approaches [14, 10] have relied on a similar trade-off between interpretability and computational complexity. In the supplementary material, we show the effect of applying temporal saliency rescaling on other datasets and provide possible optimizations.

| Saliency Methods | Middle Box | | | | Moving Box | | | |
|---|---|---|---|---|---|---|---|---|
| | AUPR | AUP | AUR | AUC | AUPR | AUP | AUR | AUC |
| Grad | 0.331 | 0.328 | 0.457 | 64.90 | 0.225 | 0.229 | 0.394 | 95.35 |
| DLS | 0.344 | 0.344 | 0.452 | 68.30 | 0.288 | 0.288 | 0.435 | 94.05 |
| SG | 0.294 | 0.300 | 0.451 | 64.00 | 0.241 | 0.247 | 0.395 | 92.90 |
| TSR + Grad | **0.399** | **0.381** | **0.471** | **62.20** | **0.335** | **0.326** | **0.456** | **84.00** |

Table 1: Results from TCN on Middle Box and Moving Box synthetic datasets. Higher AUPR, AUP, and AUR values indicate better performance. AUC lower values are better as this indicates that the rate of accuracy drop is higher.

## 8 Summary and Conclusion

In this work, we have studied deep learning interpretation methods when applied to multivariate time series data on various neural network architectures. To quantify the performance of each (interpretation method, architecture) pair, we have created a comprehensive synthetic benchmark where positions of informative features are known. We measure the quality of the generated interpretation by calculating the degradation of the trained model accuracy when inferred salient features are masked. These feature sets are then used to calculate the precision and recall for each pair.

Interestingly, we have found that commonly-used saliency methods, including both gradient-based, and perturbation-based methods, fail to produce high-quality interpretations when applied to multivariate time series data. However, they can produce accurate maps when multivariate time series are represented as either images or univariate time series. That is, when temporal and feature domains are combined in a multivariate time series, saliency methods break down in general. The exact mathematical mechanism underlying this result is an open question. Consequently, there is no clear distinction in performance between different interpretability methods on multiple evaluation metrics when applied to multivariate time series, and in many cases, the performance is similar to random saliency. Through experiments, we observe that methods generally identify salient time steps but cannot distinguish important vs. non-important features within a given time step. Building on this observation, we then propose a two-step temporal saliency rescaling approach to adapt existing saliency methods to time series data. This approach has led to substantial improvements in the quality of saliency maps produced by different methods.

# 9   Broader Impact

The challenge presented by meaningful interpretation of Deep Neural Networks (DNNs) is a technical barrier preventing their serious adoption by practitioners in fields such as Neuroscience, Medicine, and Finance [40, 41]. Accurate DNNs are not, by themselves, sufficient to be used routinely in high stakes applications such as healthcare. For example, in clinical research, one might like to ask, "why did you predict this person as more likely to develop a certain disease?" Our work aims to answer such questions. Many critical applications involve time series data, e.g., electronic health records, functional Magnetic Resonance Imaging (fMRI) data, and market data; nevertheless, the majority of research on interpretability focuses on vision and language tasks. Our work aims to interpret DNNs applied to time series data.

Having interpretable DNNs has many positive outcomes. It will help increase the transparency of these models and ease their applications in a variety of research areas. Understanding how a model makes its decisions can help guide modifications to the model to produce better and fairer results. Critically, failure to provide faithful interpretations is a severe negative outcome. Having no interpretation at all is, in many situations, better than trusting an incorrect interpretation. Therefore, we believe this study can lead to significant positive and broad impacts in different applications.

## Acknowledgements

We thank Kalinda Vathupola for his thoughtful feedback on this work. This project was supported in part by NSF CAREER AWARD 1942230, NSF award CDS&E:1854532, a grant from NIST 60NANB20D134 and HR00112090132, AWS Machine Learning Research Award and Simons Fellowship on "Foundations of Deep Learning."

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
