[Supplementary Material]

# Supplementary Material: Benchmarking Deep Learning Interpretability in Time Series Predictions

**Aya Abdelsalam Ismail, Mohamed Gunady, Héctor Corrada Bravo**[*]**, Soheil Feizi** [*]
{asalam,mgunady,sfeizi}@cs.umd.edu, hcorrada@umiacs.umd.edu
Department of Computer Science, University of Maryland

## 1 Saliency Methods

We compare popular backpropagation-based and perturbation based post-hoc saliency methods; each method provides feature importance, or "relevance", at a given time step to each input feature in a network.The relevance $R^c(x_{i,t})$ produced by saliency methods can be defined as:

- **Backpropagation-based methods**:

  - *Gradient (GRAD)*[1] the gradient of the output with respect to $x_{i,t}$:

  $$\frac{\partial S_c(X)}{\partial x_{i,t}}$$

  - *Integrated Gradients (IG)* [2] uses the average gradient while input changes from a non-informative reference point $\overline{X}$ to $X$. The relevance $R^c(x_{i,t})$ will depend upon the choice the reference point $\overline{X}$ (which is often set to zero).

  $$(x_{i,t} - \overline{x}_{t_i}) \times \int_{\alpha=0}^{1} \frac{\partial S_c\left(\overline{X} + \alpha\left(X - \overline{X}\right)\right)}{\partial x_{i,t}} d\alpha$$

  - *SmoothGrad (SG)* [3] computes the gradient $n$ times adding Gaussian noise $\mathcal{N}(0, \ \sigma^2)$ with standard deviation $\sigma$ to the input at each time.

  $$\frac{1}{n}\sum_{1}^{n} \frac{\partial S_c(X + \mathcal{N}(0, \ \sigma^2))}{\partial x_{i,t}}$$

  - *DeepLIFT (DL)* [4] a back-propagation based approach that defines a reference point and compares the activation of each neuron to its reference activation; assigning relevance according to the difference.
  - *Gradient SHAP (GS)* [5] relevance is computed by adding Gaussian noise to each input sample multiple times (similar to SmoothGrad), selects a point along the path between a reference point and input, and computes the gradient of outputs with respect to those selected points. The Shapley value is the expected value of the gradients multiplied by the difference between input and reference point.
  - *Deep SHAP (DeepLIFT + Shapley values) (DLS)* [5] Approximates the SHAP values using DeepLIFT; instead of a single reference point DeepLIFT takes a distribution of baselines computes the attribution for each input-baseline pair and averages the resulting attributions per input example; Shapley equations are used to linearize components such as max, softmax, products, divisions, etc..

---

[*]Authors contributed equally

- **Perturbation-based:**
    - *Feature Occlusion (FO)* [6] computes attribution as the difference in output after replacing each contiguous region with a given baseline. For time series we considered continuous regions as features with in same time step or multiple continuous time steps.
    - *Feature Ablation (FA)* [7] involves replacing each input feature with a given baseline, and computing the difference in output. Input features can also be grouped and ablated together rather than individually.
    - *Feature permutation (FP)* [8] randomly permutes the feature values within a batch and computes the change in output as a result of this modification. Similarly, to feature ablation input features can also be grouped and ablated together rather than individually.
- **Others:**
    - *Shapley Value Sampling (SVS)* [9] Shapley value measure the contribution of each input features by taking each permutation of the feature and adding them one-by-one to a given baseline and measuring the difference in the output after adding the features. Shapley Value Sampling is an approximation of Shapley values that involves sampling some random permutations of the input features and average the marginal contribution of features based the differences on these permutations.
    - **Random** as a control; we compare methods to a random assignment of importance.

## 2 Dataset Design

| Design Aspect | Levels | Middle Box | | Moving Box | | Positional Box | | Rare Time | | Rare Feature | | MNIST |
|---|---|---|---|---|---|---|---|---|---|---|---|---|
| | | N | S | N | S | T | F | N | M | N | M | |
| Signal Position over Time | Same | • | • | | | | • | • | | • | • | |
| | Different | | | • | • | • | | | • | | | • |
| Signal Position over Features | Same | • | • | | | • | | • | • | • | | |
| | Different | | | • | • | | • | | | | • | • |
| Signal Difference | Value | • | • | • | • | | | • | • | • | • | |
| | Position | | | | | • | • | | | | | |
| | Shape | | | | | | | | | | | • |
| Signal Abundance over Time | Abundant | • | | • | | • | • | | | • | • | • |
| | Rare | | • | | • | | | • | • | | | |
| Signal Abundance over Features | Abundant | • | | • | | • | • | • | • | | | • |
| | Rare | | • | | • | | | | | • | • | |

Figure 1: Different evaluation datasets used for benchmarking saliency methods. Some datasets have multiple variations shown as sub-levels. N/S: normal and small shapes, T/F: temporal and feature positions, M: moving shape. All datasets are trained for binary classification, except MNIST. Examples are shown above each dataset, where dark red/blue shapes represent informative features.

Figure 2: Middle box dataset generated by different time series processes. The first row shows how each feature changes over time in different samples and the bottom row corresponds to the heatmap of each sample where red represents informative features.

We design multiple synthetic datasets where we can control and examine different design aspects that emerge in typical time series datasets. Different dataset combinations are shown in Figure 1. The specific features and the time intervals (dark red/blue areas) that are considered informative is varied

between datasets to capture different scenarios of how features vary over time. As shown in Figure 1, we consider the following sub-levels:

- **Shape Normal/Small:** We modify the classification difficulty by decreasing the number of informative features. For *Middle box* and *Moving box* datasets we consider two scenarios: **Normal shape** where more than $35\%$ of overall features are informative. **Small shape** less then $10\%$ of overall features are informative.

- **Signal Normal/Moving:** The location of the importance box differs in each sample.

- **Positional Temporal/Feature:** The classification does not depend on the value of informative signal $\mu$, rather the position of informative features. **Temporal position** each class has a constant temporal position; however, the informative features in the informative temporal window change in between samples. **Feature position** each class has a constant group of features that are informative; however, the time at which these groups are informative is different between samples.

- **Rare Time/Feature:** Mimic anomalies in time series variables, identification of such deviations is important in anomaly detection tasks. **Rare Time** Most features in a small temporal window are informative; this can be static or moving, i.e., N/M. **Rare Feature** a small group features are informative in most time steps. Note that in both rare cases, less than $5\%$ of overall features are informative.

Each synthetic dataset is generated by seven different processes as shown in Figure 2. Data generation and time sampling was done in an non-uniform manner using python TimeSynth [2] package. The base time series were generation by the following processes note that $\varepsilon_t \sim \mathcal{N}(0,1)$

- Gaussian noise with zero mean and unit variance.
$$X_t = \varepsilon_t$$

- Independent sequences sampled from a harmonic function. A sinusoidal wave was used with $f = 2$.
$$X(t) = \sin(2\pi f t) + \varepsilon_t$$

- Independent sequences sampled from a pseudo period function, where, $A_t \sim \mathcal{N}(0, 0.5)$ and $f_t \sim \mathcal{N}(2, 0.01)$
$$X(t) = A_t \sin(2\pi f_t t) + \varepsilon_t$$

- Independent sequences of an autoregressive time series process, where, $p = 1$ and $\varphi = 0.9$
$$X_t = \sum_{i=1}^{p} \varphi_i X_{t-i} + \varepsilon_t$$

- Independent sequences of a continuous autoregressive time series process, where, $\varphi = 0.9$ and $\sigma = 0.1$.
$$X_t = \varphi X_{t-1} + \sigma(1 - \varphi)^2 * \varepsilon + \varepsilon_t$$

- Independent sequences of non–linear autoregressive moving average (NARMA) time series, where, the equation is given below, where $n = 10$ and $U \sim U(0, 0.5)$ is a uniform distribution.

$$X_t = 0.3X_{t-1} + 0.05X_{t-1} \sum_{i=0}^{n-1} X_{t-i} + 1.5U\left(t - (n-1)\right) * U(t) + 0.1 + \varepsilon_t$$

- Independent sequences sampled according to a Gaussian Process mixture model with selected covariance function [10].

Informative features are then highlighted by the addition of a constant $\mu$ to positive class and subtraction of $\mu$ from negative class (unless specified, $\mu = 1$).

**Multivariate MNIST time series** is included as a more general case, each sample has 28 time steps, and the feature embedding size is 28.

# 3 Saliency Distribution

**Real Data: Human Connectome Project fMRI Data:**

To inspect saliency distribution in a more realistic setting, we apply different saliency methods and plot the distribution of ranked features on an openly available fMRI dataset of the Human Connectome Project (HCP) [11]. In this dataset, subjects are performing specific tasks while scanned by an fMRI machine. Our classification problem is to identify the task performed, given the fMRI scans. The distribution of different saliency methods across multiple neural architectures is shown in Figure 3. Similar to synthetic data, saliency exponentially decays with feature ranking.

Figure 3: The distribution of saliency values of ranked features produced by different saliency methods for HCP fMRI data.

**Time Series MNIST:**

Figure 4: The distribution of saliency values of ranked features produced by different saliency methods for Time Series MNIST.

**Synthetic Data:**

Figure 5 shows the saliency distribution for different *(neural architecture, saliency method)* pairs. Aside from feature ablation, saliency decays exponentially with feature ranking, the distribution across different methods and datasets seem to be similar for a neural architecture.

# 4 Performance Evaluation Metrics

Given the synthetic data described earlier, informative features are known (dark areas in Figure 1), and we can calculate precision and recall of each *(neural architecture, saliency method)* pair using the confusion matrix in Table 1.

| Actual / Saliency | Informative | Noise |
|---|---|---|
| High | True Positive (TP) | False Positive (FP) |
| Low | False Negative (FN) | True Negative (TN) |

Table 1: Confusion Matrix, for precision and recall calculation.

Figure 5: The distribution of saliency values of ranked features produced by different saliency methods for various synthetic datasets. Empty spaces indicates that model was not able to learn classification task for the given dataset.

**Precision**

The fraction of informative high saliency features among all high saliency features. Since the saliency value varies dramatically across features, we do not look at the number of true positive and false negative instead their saliency value; the (weighted) precision is calculated by:

$$\frac{\sum R\left(x_{t_i}\right)\left\{x_{t_i} : x_{t_i} \in TP\right\}}{\sum R\left(x_{t_i}\right)\left\{x_{t_i} : x_{t_i} \in TP\right\} + \sum R\left(x_{t_i}\right)\left\{x_{t_i} : x_{t_i} \in FP\right\}}$$

**Recall**

The fraction of the total informative features that had high saliency, similar to the precision we use the saliency value rather than the count. (Weighted) the recall is defined as:

$$\frac{\sum R\left(x_{t_i}\right)\left\{x_{t_i} : x_{t_i} \in TP\right\}}{\sum R\left(x_{t_i}\right)\left\{x_{t_i} : x_{t_i} \in TP\right\} + \sum R\left(x_{t_i}\right)\left\{x_{t_i} : x_{t_i} \in FN\right\}}$$

Through our experiments, we report area under the precision curve (AUP), the area under the recall curve (AUR), and area under precision and recall (AUPR). The curves are calculated by the precision/recall values at different levels of degradation. We also consider feature/time precision and recall (a feature is considered informative if it has information at any time step and vice versa). For the random baseline, we stochastically select a saliency method then permute the saliency values producing arbitrary ranking.

## 5    Temporal Saliency Rescaling Optimizations and Complexity

The main back draw of **T**emporal **S**aliency **R**escaling (Algorithm 1) is the increase in complexity that is a result of performing multiple gradient calculations. Algorithm 2 shows a variation of the algorithm that calculates the contribution of a group of features within a time step. Algorithm 3 calculates the contribution of each time step and feature independently; the total contribution of a single feature at a given time is the product of feature and time contributions.

---

**Algorithm 1:** Temporal Saliency Rescaling (TSR)

---

**Given:**  input $X$, a baseline interpretation method $R(.)$
**Output:** TSR interpretation method $R^{TSR}(.)$
**for** $t \leftarrow 0$ **to** $T$ **do**
  Mask all features at time $t$: $\overline{X}_{:,t} = 0$, otherwise $\overline{X} = X$;
  Compute Time-Relevance Score $\Delta_t^{time} = \sum_{i,t} |R_{i,t}(X) - R_{i,t}(\overline{X})|$;
**for** $t \leftarrow 0$ **to** $T$ **do**
  **for** $i \leftarrow 0$ **to** $N$ **do**
    **if** $\Delta_t^{time} > \alpha$ **then**
      Mask feature $i$ at time $t$: $\overline{X}_{i,t} = 0$, otherwise $\overline{X} = X$;
      Compute Feature-Relevance Score $\Delta_i^{feature} = \sum_{i,t} |R_{i,t}(X) - R_{i,t}(\overline{X})|$;
    **else**
      Feature-Relevance Score $\Delta_i^{feature} = 0$;
    Compute (time,feature) importance score $R_{i,t}^{TSR} = \Delta_i^{feature} \times \Delta_t^{time}$ ;

---

The approximate relevance calculations needed for each variation is shown in table 2. The complexity **TSR** and **TSR With Feature Grouping** highly depends on $\alpha$. In many time series applications such as anomaly detection, $\alpha$ can be set to be close to 1. **TFSR** complexity is comparable to SmoothGrad. Other approaches have proposed similar trade-offs between interpretability and computational complexity, i.e., Hooker et al. [12] proposed retraining the entire network after removing salient features, retraining even most simple networks is very expensive in comparison to extra gradient calculations.

---

**Algorithm 2:** Temporal Saliency Rescaling (TSR) With Feature Grouping

---

**Given:** input $X$, a baseline interpretation method $R(.)$, feature group size $G$
**Output:** TSR interpretation method $R^{TSR+FG}(.)$
**for** $t \leftarrow 0$ **to** $T$ **do**
  Mask all features at time $t$: $\overline{X}_{:,t} = 0$, otherwise $\overline{X} = X$;
  Compute Time-Relevance Score $\Delta_t^{time} = \sum_{i,t} |R_{i,t}(X) - R_{i,t}(\overline{X})|$;

**for** $t \leftarrow 0$ **to** $T$ **do**
  **for** $i \leftarrow 0, G, 2G, \ldots, N$ **do**
    **if** $\Delta_t^{time} > \alpha$ **then**
      Mask features $i : i + G$ at time $t$: $\overline{X}_{i:i+G,t} = 0$, otherwise $\overline{X} = X$;
      Compute Feature-Relevance Score $\Delta_{i:i+G}^{feature} = \sum_{i,t} |R_{i,t}(X) - R_{i,t}(\overline{X})|$;
    **else**
      Feature-Relevance Score $\Delta_{i:i+G}^{feature} = 0$;
    Compute (time,feature) importance score $R_{i,t}^{TSR+FG} = \Delta_{i:i+G}^{feature} \times \Delta_t^{time}$ ;

---

---

**Algorithm 3:** Temporal Feature Saliency Rescaling (TFSR)

---

**Given:** input $X$, a baseline interpretation method $R(.)$
**Output:** TFSR interpretation method $R^{TFSR}(.)$
**for** $t \leftarrow 0$ **to** $T$ **do**
  Mask all features at time $t$: $\overline{X}_{:,t} = 0$, otherwise $\overline{X} = X$;
  Compute Time-Relevance Score $\Delta_t^{time} = \sum_{i,t} |R_{i,t}(X) - R_{i,t}(\overline{X})|$;

**for** $i \leftarrow 0$ **to** $N$ **do**
  Mask all time steps for feature $i$: $\overline{X}_{i,:} = 0$, otherwise $\overline{X} = X$;
  Compute Feature-Relevance Score $\Delta_i^{feature} = \sum_{i,t} |R_{i,t}(X) - R_{i,t}(\overline{X})|$;

**for** $t \leftarrow 0$ **to** $T$ **do**
  **for** $i \leftarrow 0$ **to** $N$ **do**
    Compute (time,feature) importance score $R_{i,t}^{TFSR} = \Delta_i^{feature} \times \Delta_t^{time}$ ;

---

| Algorithm | Approximate number of Relevance Calculations |
|---|---|
| Algorithm 1: $R^{TSR}(.)$ | $T + (T * (1 - \alpha) * N)$ |
| Algorithm 2: $R^{TSR+FG}(.)$ | $T + (T * (1 - \alpha) * N/G)$ |
| Algorithm 3: $R^{TFSR}(.)$ | $T + N$ |

Table 2: Complexity analysis of different varaitions of **TSR**

# 6 Experiments and Results

**Saliency Map Quality**

From figures 6, 7 and 8, when applying Temporal Saliency Rescaling we observe a definite improvement in saliency quality across different architectures and interpretability methods except for Gradient SHAP and SmoothGrad.

**MNIST**

Figure 6 shows saliency maps produced by each *(neural architecture, saliency method)* pair on samples from time series MNIST; Figure 7, show the samples after applying **TSR**. There is a significant improvement in the quality of the saliency map after applying the Temporal Saliency Rescaling approach.

Figure 6: Saliency maps produced by Gradient-based saliency methods including Grad, Integrated Gradients, DeepLIFT, Gradient SHAP, DeepSHAP and SmoothGrad and non-gradient-based saliency method including Shap value sampling, Feature Ablation and Feature Occlusion for 4 different models on time series MNIST (white represents high saliency).

Figure 7: Saliency maps when applying the proposed Temporal Saliency Rescaling (TSR) approach on different saliency methods.

**Synthetic Datasets**

Figure 8 shows saliency maps produced by each *(neural architecture, saliency method)* pair on samples from different synthetic datasets before and after applying **TSR**.

**(a)**   **(b)**

Figure 8: Saliency maps produced by Grad, Integrated Gradients, DeepLIFT, Gradient SHAP, DeepSHAP, and SmoothGrad for three different models on static synthetic datasets. (b)Saliency maps when applying the proposed Temporal Saliency Rescaling (TSR) approach.

## Saliency Methods versus Random Ranking

### Model Accuracy Drop, Precision and Recall

The effect of masking salient features on the model accuracy is shown in the first row of Figures [9-18]. Similarly, precision and recall at different levels of degradtion are shown in second row of Figures [9-18].

Figure 9: Accuracy drop, precision and recall for ***Middle box*** datasets

Figure 10: Accuracy drop, precision and recall for ***Small Middle box*** datasets

Figure 11: Accuracy drop, precision and recall for ***Moving Middle box*** datasets

Figure 12: Accuracy drop, precision and recall for ***Small Moving Middle box*** datasets

Figure 13: Accuracy drop, precision and recall for *Rare Feature* datasets

Figure 14: Accuracy drop, precision and recall for **Moving Rare Feature** datasets

Rare Time

Figure 15: Accuracy drop, precision and recall for **_Rare Time_** datasets

Figure 16: Accuracy drop, precision and recall for *Moving Rare Time* datasets

Figure 17: Accuracy drop, precision and recall for *Positional Feature* datasets

Figure 18: Accuracy drop, precision and recall for *Positional Time* datasets

**Saliency Maps for Images versus Multivariate Time Series**

Figure 19 shows a few examples of saliency maps produced by the various treatment approaches of the same sample (images for CNN, uni, bi, multivariate time series for TCN). One can see that CNN and univariate TCN produce interpretable maps. In contrast, the maps for the bivariate and multivariate Grad are harder to interpret, applying the proposed Temporal Saliency Rescaling approach on bivariate and multivariate time series significantly improves the quality of saliency maps and in some cases even better than images or univariate time series.

Figure 19: Saliency Maps for samples when treated as an image (CNN) vs. univariate (1 feature x 784 time steps), bivariate (2 features x 392 time steps), or multivariate (28 features x 28 time steps) time series (TCN) before and after applying TSR.

## Footnotes

[2]https://github.com/TimeSynth/TimeSynth