[Reviews · NeurIPS 2020]

Review 1

Summary and Contributions: This paper introduces a set of benchmark datasets for timeseries saliency methods, a series of associated metrics for evaluating timeseries saliency methods, an empirical conclusion that common methods produce poor saliency maps on timeseries data, and a method for improving them.

Strengths: - The empirical evaluation seems very solid. - The comparison with CNNs makes it convincing that the saliency problems they identify really are specific to timeseries models, which seems like a novel finding. - Research on evaluating saliency methods with respect to objective ground-truth is always nice to see and has generally been more significant than interpretability papers which are only grounded subjectively. - Overall the problem is relevant.

Weaknesses: - As the authors acknowledge, they do not provide much theoretical insight into why such problems occur with saliency methods on timeseries data, or why certain methods (e.g. LSTMs) are generally "harder to interpret" - The paper relies entirely on gradient-based saliency methods (even for SHAP approximation), but there are more diverse/lower tech approaches, and leaving them out makes the paper less informative. For example, I'd be curious to see an empirical comparison to simple masking-based saliency (i.e. changes in model predictions when a feature is set to 0 at a timestep). Perhaps that would do comparatively better without TSR and would provide insight into why TSR is helpful.

Correctness: Overall, the claims and methodology seem sound.

Clarity: The paper is well and clearly written. However, the figures are definitely too small -- at least the ones which were squeezed into the main paper, rather than the supplement :)

Relation to Prior Work: Although I am not extremely well versed in the time-series literature, it seems like the paper does a good job discussing its relationship to the current state of the field.

Reproducibility: Yes

Additional Feedback: Update in response to author feedback: I still feel this paper is good and worth including, despite other reviewers' concerns that the contribution is too marginal. I also think the benchmark datasets this paper introduces are particularly valuable.


Review 2

Summary and Contributions: This submission compares the performance of various saliency-based interpretability methods across diverse neural architectures on a set of synthetic datasets. The authors also propose a simple two-step temporal saliency rescaling approach to improve the performance for time series data. Overall, this work is OK, but lacks in-depth analysis and novelty.

Strengths: It is a meaningful work to establish a test data set to compare saliency-based interpretability methods for time series data. The new simple method proposed in the submission also have a significant performance improvement on this synthetic dataset.

Weaknesses: This submission lacks depth and novelty, and has limited contributions to the machine learning community. The tested data set is an extension of previous synthetic data. It is a question to me whether this data set covers most issues real data has. This paper also lacks a theoretical analysis of why the previous metrics failed and why their proposed metric have better performance. There is no in-depth discussion of the limitations of the new metric. In addition, there is a lack of guidance on the selection of the threshold in the algorithm.

Correctness: Yes, I think the proposed method is reasonable.

Clarity: Yes, this submission is well written and well organized

Relation to Prior Work: Yes, the authors clearly discussed the related works in the Background and Related Work section.

Reproducibility: Yes

Additional Feedback: Thank you for authors to address my concerns in the rebuttal. I remain my score unchanged because I still think the novelty of this submission is not enough for a NeurIPS publication.


Review 3

Summary and Contributions: This paper is an extensive evaluation of performance of saliency and gradient based methods (developed extensively for images) for time-series (classification) interpretability. The time-series setup is interesting because importance of features can change over time for the classification setting. The claim of the paper is that saliency methods do not extend well to time series. Based on this observation, the authors propose a new algorithm that first finds important times by masking the time and evaluating change in relevance score and then obtaining feature importance at relevant time points. Experimental evaluation shows that among the class of saliency methods, the authors' proposed method provides better interpretability for evaluated datasets.

Strengths: The experimental evaluation is quite extensive. The work is relevant to the NeurIPS audience. I have some questions regarding empirical evaluation which I have outlined in the following. If not addressed, I am uncertain of the soundness of the claims as well as diminishes the novelty of the contribution to some extent. I did see some useful discussions around the choice of evaluation metrics for saliency methods.

Weaknesses: 1. My first concern is that the setup is not explained clearly. Problem Definition should be more elaborate. Is it a time-series classification setting? Will the conclusions apply if the label is available for every time instance t? Are all time points explaining the prediction y at t=T? That is, what is S(X) representing? 2. I think the premise that gradient based saliency methods might work in time-series settings is valid, but I am not entirely convinced why they will work knowing that RNNs, LSTMs, suffer from the vanishing gradient problem, which seems to have inspired the work of [21] Aya Abdelsalam Ismail, Mohamed Gunady, Luiz Pessoa, Hector Corrada Bravo, and Soheil Feizi. Input-cell attention reduces vanishing saliency of recurrent neural networks. In Advances in Neural Information Processing Systems 32, 2019. Nonetheless there is some merit in exhaustively evaluating existing methods on time-series data. Which brings me to my concern around how such datasets were designed. 3. 6 out of 7 datasets used for this extensive evaluation are not time-series data, but image data where one spatial axis is assumed to be temporal. I have major concerns around evaluating time-series data this way. It is highly unclear (neither is it justified in the paper) why this is a reasonable evaluation setup. Why weren't temporal generative process, state-space models, even gaussian process used as reliable data-generating mechanisms on which these methods are evaluated? The main insight would still likely be that saliency methods are noisy and architectural challenges of RNNs around vanishing saliency will in-fact affect the quality of the explanations. The authors finally mention this in Line 226: "These observations suggest that saliency maps fail when feature and time domains are conflated". I think reaching this conclusion doesn't say much about the methods at all and is more a claim of the data-set generation and assumption. 4. Authors claim in introduction that they have compared SHAP method but I only see SHAP+Gradient derivates rather than the original SHAP method. This method is not developed for time-series, only for tabular data, so it is unclear how they would do this evaluation, neither is it described anywhere in the main draft.

Correctness: It will help if the authors clarify all the concerns I raised above. As of now I do not believe the methodology is entirely correct. The claim that saliency methods cannot identify important features but identify important times needs to be tested on realistic time-series rather than images where one axis is assumed temporal. The correlations in temporal settings are hard to conflate with such image based correlations.

Clarity: The paper is well written and all conclusions clearly stated.

Relation to Prior Work: Yes prior work is clearly outlined, motivation of the study is clear, all baseline descriptions are clear.

Reproducibility: Yes

Additional Feedback: I have read the author response and acknowledge that they have taken a lot of care to address the concerns I have raised. With that in mind, I have raised the score from 3->4. I suggest authors include other attribution baselines relevant to time-series in their evaluation (other than standard saliency ones to form a convincing argument of their two-step saliency approach). ============================================================ Please see above for all concerns raised.


Review 4

Summary and Contributions: The main contributions of this paper are: 1- Presenting an extensive study and analysis of existing saliency-based interpretability methods on temporal data. 2- Proposing a two-step temporal saliency rescaling (TSR) approach.

Strengths: Strengths are mentioned in the previous section. Besides, the proposed method performs better than existing ones.

Weaknesses: Authors did a great job of studying and analysing the previous methods. But the proposed method is expensive and also not novel.

Correctness: Yes, claims and method are sound and correct.

Clarity: This paper is well-written.

Relation to Prior Work: Yes. Prior works are discussed and the difference is identified properly.

Reproducibility: Yes

Additional Feedback:

[Author Response · NeurIPS 2020]

**Response to Reviewer 1:** - You are correct that "comparison with CNNs makes it convincing that the saliency problems" addressed in our paper are specific to multi-variate "timeseries models". This finding is the key message of our paper. Even though the mathematical explanation of that behavior remains an open question, we focused on identifying the problem, benchmarking that behavior across different data characteristics using objective ground truth, and proposing a rescaling approach that enhances the quality of time-series saliency methods. - Upon your suggestion, we repeat some experiments by masking features and evaluating changes in the loss (see AUPR of some datasets in the table below). We find that although Masking is very expensive, it does not perform well.

| Methods | Middle Box | | | Rare Time | | |
|---|---|---|---|---|---|---|
| | LSTM+In. | TCN | Trans. | LSTM+In. | TCN | Trans. |
| Grad | 0.494 | 0.237 | 0.288 | 0.218 | **0.334** | 0.027 |
| SG | 0.427 | 0.312 | 0.295 | 0.025 | 0.134 | 0.012 |
| Masking | 0.332 | 0.348 | 0.382 | 0.180 | 0.000 | 0.028 |
| TSR+Grad | **0.677** | **0.541** | **0.437** | **0.378** | 0.103 | **0.307** |

Table 1: AUPR: With feature masking

Figure 1: Experiments on new NARMA dataset

**Response to Reviewer 2: Novelty and contributions.** In our paper, we make following contributions: (1) We identify a limitation in the saliency methods when applied to multi-variate timeseries models, this finding itself is novel. (2) We show that this limitation exists in different neural architectures and across a large range of popular saliency methods in different setups. (3) We create benchmark metrics to evaluate models/methods that can be used on datasets where informative features are known, we show that methods should be compared based on precision and recall of features identified as salient not only with model accuracy drop upon elimination as in standard practice. (4) We propose a novel method TSR that improves the quality of saliency methods for timeseries. We believe these contributions can be of interest to the ML community.

**Richness of tested datasets.** The benchmark itself was done on synthetic data to have a ground truth to quantify performance of different methods. We have created a wide variety of datasets to cover important aspects of time series data in a classification set up. To show easy-to-interpret saliency maps in addition to the quantified metrics, we used timeseries MNIST [1] over which we demonstrated issues of saliency maps and showed how TSR improves it. To verify that our metrics are reasonable, we calculated saliency distributions on real fMRI time series dataset (supplementary page 3) where we observe similar trends as in synthetic datasets.

**Limitations of the new metric & guidance on threshold selection.** The limitation of metrics is the need for ground truth, hence the need for synthetic data. Limitation of TSR is cost and depending on the $\alpha$ as an optimization choice that controls the time complexity of TSR (supplementary page 9&10). We will include a more detailed discussion.

**Response to Reviewer 3: Problem setup:** It is a time series classification setup where all timesteps contribute to making the final output, labels are available after the last timestep (this setup is used in many real-case applications such as an fMRI task classification problem and we are interested in seeing how feature importance change across time[2]). $S(X)$ is network output i.e. $S_1(X)$ is the probability of class 1.

**Using gradient-based saliency methods with architectures suffering from vanishing gradients:** 3 out of 4 architectures (TCN, Transformer and Input-cell attention proposed by Ismail et al) tested in the benchmark do not suffer from vanishing gradient problem. As discussed, we find that the gradient-based saliency methods limitations for multivariate timeseries is consistent across such architectures in different setups.

**Regarding evaluation datasets:** Upon your suggestion, we generated a new dataset with the advised setting. Informative feature in new dataset follows a non–linear autoregressive moving average (NARMA) and the non-informative features are Gaussian processes with mean and standard deviation uniformly chosen. The class assignment is generated as the sign of summation of salient features at different informative timesteps. As shown in figure 1, findings are consistent with that of the main paper; the behavior of temporal generated datasets are similar to paper's synthetic datasets. We will include this dataset along with more datasets generated using HMMs and state models in the benchmark.

**Use of GradientSHAP for timeseries:** GradientShap approximates SHAP values by computing the expectations of gradients by randomly sampling from the distribution of baselines/references. Like others [3,4], we believe such method can be used for timeseries data. We will consider replacing GradientSHAP with SHAP in our final draft.

**Response to Reviewer 4:** Regarding the proposed approach (TSR). It is an early attempt to tackle the problem observed in all the benchmarked saliency methods which fail to highlight important features within a time step. Our approach shows that TSR improves the quality of the produced saliency maps. However, a more elaborate solution would be essential as soon as a more extensive understanding of the fundamental causes of that problem is achieved. We present this open problem to the community in this paper along with a ready-to-use benchmark with which to approach it.

[1] Bai, et al. "An empirical evaluation of generic convolutional and recurrent networks for sequence modeling".
[2] Thomas, Armin W., et al. "Interpretable LSTMs for whole-brain neuroimaging analyses."
[3] Tonekaboni, Sana, et al. "What went wrong and when? Instance-wise Feature Importance for Time-series Models."
[4] Lundberg, Scott, et al."Explainable machine-learning predictions for the prevention of hypoxaemia during surgery.".


[Meta-Review · NeurIPS 2020]

This work introduces a bunch of benchmarks for evaluating time series saliency methods (with respective metrics). The authors do a number of empirical evaluations, draw some conclusions about why certain things don't work, and propose a new saliency method based on that. There are a number of things that I like about this work and that was pointed out by the reviewers as well: there is a definite lack of datasets with groundtruth saliency in them so coming up with such a dataset (and associated metrics) is a worthy contribution by itself (though perhaps not rising up to the bar of acceptance at NeurIPS). In general, everyone agreed that this part of the paper is good. What was more controversial: is the subsequent analysis interesting and novel enough? I can see the arguments on both sides of this. On the one hand, the reliance on mostly time series of *image* data is somewhat limiting in my opinion. I understand that the rebuttal proposes a new dataset, but it's unclear to me how relevant that dataset is. There's also some discussion (in the rebuttal period) about whether the two-step approach is actually that novel or useful for time-series and whether the experiments show this. There's merit to the criticism that the synthetic results don't necessarily translate well into conclusions on real-world datasets. On the other hand, I do think that the idea of using these synthetic benchmarks as a sort of a "unit test" could be appealing. And the authors seemed to have made a genuine effort with their fMRI results to show real-world relevance. All in all, I believe this work could appeal to NeurIPS audience and enrich the literature on this topic, especially if the authors heed the advice of adding more bona-fide interpretability methods (rather than just saliency) in the camera ready version, make more of an effort to understand how this analysis applies to more realistic datasets (again, in the camera ready) and finally convince the readers of the value of TSR (via extensive ablations if appropriate).